# Lessons Learnt from the Revitalisation of Chemical Factory in Marktredwitz and River Banks Downstream: When 'Renaturation' Can Be Harmful

Tomáš Matys Grygar [1,2,*], Michal Hošek [1,2], Tomáš Navrátil [3], Jan Bednárek [4], Jindřich Hönig [5], Jitka Elznicová [2], Jan Pacina [2], Jan Rohovec [3], Jan Sedláček [6] and Oliver Sass [7]

1. Institute of Inorganic Chemistry, Czech Academy of Sciences, 250 68 Řež, Czech Republic
2. Faculty of Environment, J. E. Purkyně University in Ústí nad Labem, Pasteurova 3632/15, 400 96 Ústí nad Labem, Czech Republic
3. Institute of Geology, Czech Academy of Sciences, Rozvojová 269, 165 00 Praha 6-Lysolaje, Czech Republic
4. Povodí Ohře, State Enterprise, Novosedlická 758, 415 01 Teplice, Czech Republic
5. Povodí Ohře, State Enterprise, Bezručova 4219, 430 03 Chomutov, Czech Republic
6. Faculty of Science, Palacký University, 17. Listopadu 1192/12, 779 00 Olomouc, Czech Republic
7. Department of Geosciences, University of Bayreuth, Universitätsstr. 30, D-95447 Bayreuth, Germany
* Correspondence: grygar@iic.cas.cz

**Abstract:** Our study addressed mercury contamination hotspots that originated from Chemical Factory Marktredwitz, Germany. The factory was abandoned in 1985 but its legacy has been persistently endangering the river ecosystem of the Ohře River, a Labe (Elbe) River tributary in the Czech Republic. We identified the timing for the peak contamination of fine sediments entering the Skalka Reservoir located on the Ohře River downstream of the Czech German boundary. Age constraints for the reservoir sediments were obtained using gamma spectrometry analyses of $^{137}$Cs and unsupported (excess) $^{210}$Pb. We also summarised historical and current Hg concentrations in suspended particulate matter in the Kössein–Röslau–Ohře river system and recent Hg concentrations in aquatic plants. Secondary contamination and its transfer to the Czech stretch of the Ohře River and the Skalka Reservoir through severely contaminated suspended material peaked during the period of factory closure and the start of remediation. The Hg contamination import to the Czech Republic is not likely to improve if the river is left without traditional management of bank reinforcement. This case study highlights a gap in safety regulations for the management of severely contaminated rivers and demonstrates the need to consider the role of historical contamination in river 'renaturation'.

**Keywords:** contaminated rivers; embankment; revitalisation; renaturation; mercury

## 1. Introduction

Historical contamination related to the early stages of industrialisation is a current threat to biota and humans [1–3]. Researchers have conducted the mapping of historical contamination of river systems, while state authorities focus mainly on actual primary contamination or the real danger of acute toxicity to residents [1]. In recent decades, there have been attempts to establish intervention limits for risk element concentrations in soils and water, such as Dutch pollution standards, which would prompt remediation actions at the national level, even in the case of historical contamination; however, no such limits have yet been built in the European national legislation. The Water Framework Directive of the European Union (WFD, 2000/60/EU) appeared to be a step in the right direction, at least to identify local problems in each river system and propose ways to improve their actual condition. However, the WFD has provided neither general regional thresholds for river status [4] nor obligatory intervention or remediation targets. Handling historically contaminated areas can be motivated and evaluated by so many diverse aims, that the identification of societal priorities should first be addressed in planning future actions [3].

These frames are also missing for severely contaminated river systems. Little attention has been paid for minimising the risks associated with the renaturation (or lack of management) of such contaminated rivers. Historical engineering structures, which are considered obsolete, could prevent physical erosion of contaminated river banks [5–7]; however, their benefits are not always taken into account for the management of current river systems, particularly in light of WFD and current societal attitudes toward the environment.

The consequences of the missing frames, which would make life with historical industrial legacy in rivers safe, can be documented for the Kössein–Röslau–Ohře river system in eastern Bavaria, Germany and western Czech Republic (Figure 1). Contamination originated from Chemische Fabrik Marktredwitz (CFM) in the eastern Bavaria, Germany, which produced chemicals containing Hg and other risk elements during 1788–1985 [8–11]. Contamination of the Kössein River by soluble species peaked in the 1980s, and the leakage of wastewater effluents in the 1980s was the ultimate reason for factory closure [5,9,12]. The CFM was demolished, and the entire area was remediated in the 1990s [5,8]. Remediation was considered successful in the CFM area [5,8,11,12]; however, the downstream river system remained severely and persistently contaminated [2,10,12]. The Hg concentrations in the sediments of the downstream river systems are comparable to those in major European Hg mines in Almadén [13], the Monte Amiata ore region [14,15], and the Idrija catchment [16,17]. Hošek et al. [2] estimated that approximately 22 t Hg is present in the contaminated channel belt of the Kössein and Röslau rivers and hypothesised that this load will stepwise be remobilised or leached and transported downstream to the transboundary Ohře–Labe (Elbe) river system and to the Skalka Reservoir farther downstream.

Reservoirs accumulate fine particles and many chemical contaminants more efficiently and persistently than river channels and floodplains [18–22]. The Skalka Reservoir has already accumulated sufficient Hg to render predatory fish unconsumable [2,9]. Unfortunately, the contamination is continuing to spread. Accumulation in the reservoir will inevitably result in secondary contamination after reservoir filling with sediments [23] or reservoir drainage [22]. Currently, the Skalka Reservoir cannot completely prevent the export of the CFM contamination downstream to the Ohře River [19], and the Hg-contaminated sediments will end up back in Germany via the Labe (Elbe) River, wherein Hg is a prominent chemical contaminant [24].

The major goal of our study was to find time period of the most severe contamination of the fine sediments downstream of the CFM using (1) the results of long-term monitoring of the contamination parameters of the Röslau River and (2) the records in the Skalka Reservoir sedimentary archive. Contamination climax in sediments might be expected to occur simultaneously with atmospheric emissions, which peaked in the 1920s and the 1970s [11], or just before the closure of CFM in the 1980s. However, monitoring of suspended particulate matter (SPM) in the lower reach of the Röslau River revealed the increase in Hg concentrations in the late 1990s [2,9], which was the period after the closure of CFM. This timing needed verification because it might document the need for greater care in the remediation and revitalisation of contamination hotspots. The major aim of our study was to highlight the need of managing river systems with legacy contamination. There is a considerable gap in research and legislative related to revitalisation of contaminated fluvial systems; this manuscript aimed to fill the gap in research and substantiate the need to improve legislative measures.

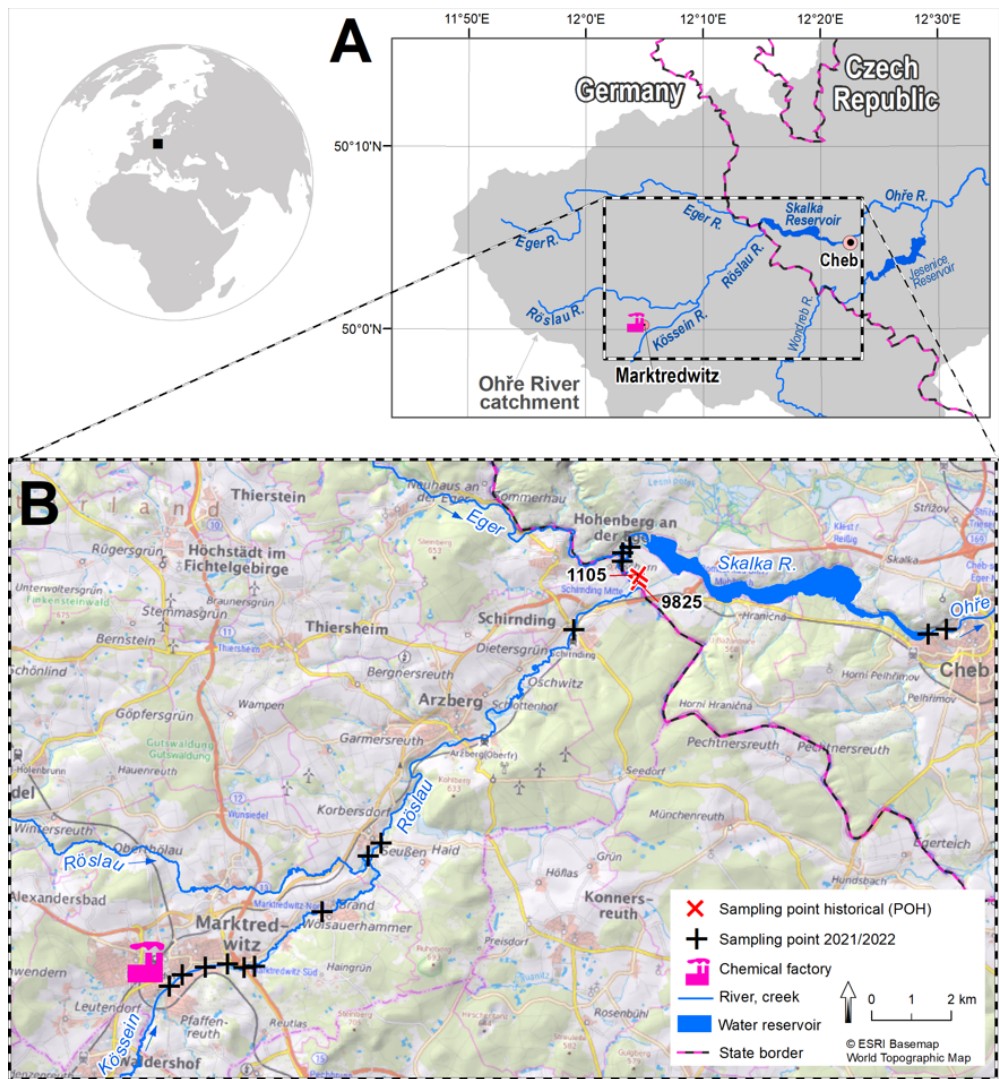

**Figure 1.** Map of the Ohře River catchment (**A**) and detailed map of the study area with position of sampling sites (**B**).

## 2. Materials and Methods

### 2.1. Study Area

The Skalka Dam in the Czech Republic was built in 1962–1964, with the original purpose of providing drinking water. The reservoir area is 3.4 km$^2$ and the maximum water depth is 12 m. The reservoir water table regularly decreases (over a period of several years) to prevent eutrophication and algal bloom [23]. The catchment area of the Ohře River upstream of the reservoir was 692 km$^2$. Water was provided by the Eger and Röslau rivers in Germany (Figure 1).

The CFM was situated at banks of the Kössein River, a tributary of the Röslau River in Bavaria. The remediation of the severely Hg-contaminated company site, starting in the mid-1980s and finishing in the mid-1990s, was the largest and most expensive redevelopment project (Sanierungsprojekt in German) in Bavaria. The most important decontamination measure in the rivers was the digging and suction of sludge from local reservoirs around 1995. However, the Kössein–Röslau river channel belt is still severely contaminated with Hg concentrations of up to several hundred mg kg$^{-1}$ [2,10]. Contaminated sediments mainly occur along the river course with a thickness of up to 1–2 m. A contaminated but thinner surface layer is also found in the floodplains, where it gets progressively buried by younger, less contaminated deposits (20–30 mg kg$^{-1}$) [5]. Mercury still entering the Skalka Reservoir is mainly derived from bank erosion during floods and is predominately

present in the suspended load. Due to this contamination input, Hg concentrations in SPM in inflow to and outflow from the Skalka Reservoir were ca. 10 and 3 mg kg$^{-1}$, respectively, in the 2010s and in the present as well [2,10].

In terms of renaturation of the Kössein and Röslau rivers, there is tension between EU nature conservation directives, in particular WFD management plans to improve river conditions and protection of Fauna–Flora–Habitat in the frame of the Natura 2000 regions, and the justified demand mainly from Czech side to keep contaminated river banks stable to prevent downstream contamination. Since the end of the remediation in 1996, sporadic damage to the old riverbank reinforcements occurred by toppling old trees, and occasional construction measures (wooden structures to prevent bank erosion) were implemented. Recently, a feasibility study on renaturation was completed, with the pivotal aim of enhancing river conditions without mobilising pollutants. The first pilot measure was implemented on a river length of approximately 400 m, including stone groynes structuring the riverbed, preventing depth erosion and bank reinforcement for erosion protection using near-natural structures [25].

### 2.2. Skalka Reservoir Sediment Sampling

Sampling in the Skalka Reservoir was performed in 2–3 October 2018 using a gravity corer (UWITEC GmbH, Mondsee, Austria) as previously described [21]. Sediments were pushed from Plexiglas cores by a piston and sliced to 1–3 cm segments. Contamination by risk elements in the river floodplains and the Skalka Reservoir revealed joint contamination by Cu, Hg, Pb, and Zn, with the onset of Hg contamination occurring little later than the onset of other risk elements [2,26]. Álvarez-Vázquez et al. [27] concluded that the Skalka Reservoir was too shallow to prevent sediment reworking by currents and wind. Therefore, we selected cores from the deepest part of the reservoir (Figure 2) and expected the possible stratigraphic disorders in the sediments.

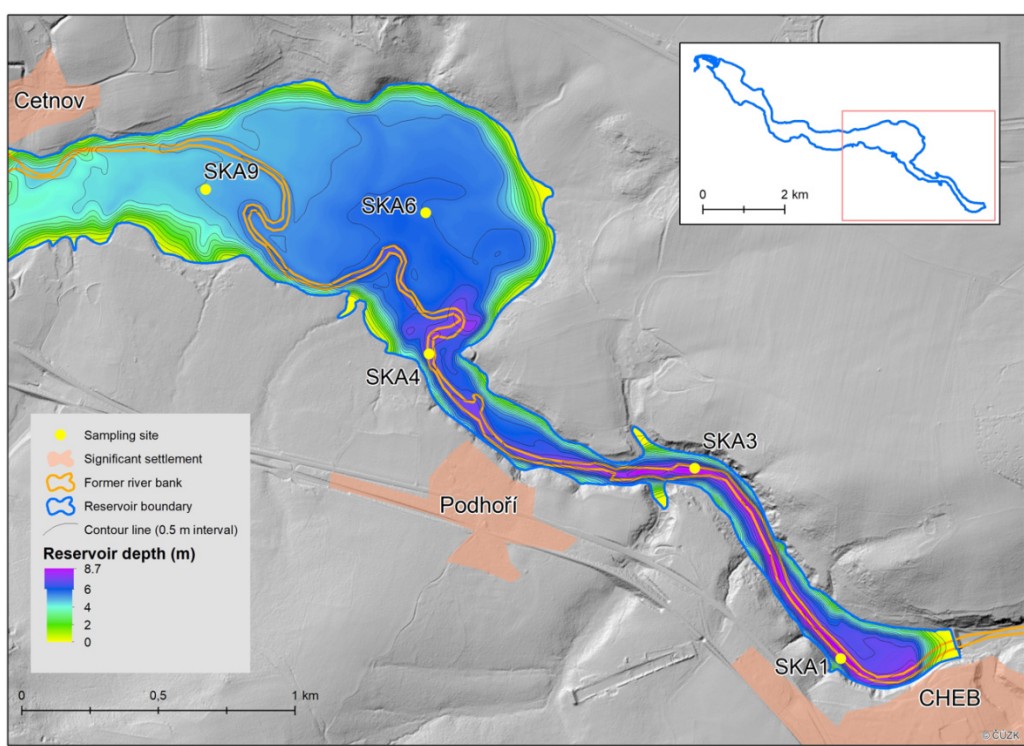

**Figure 2.** Positions of studied gravity cores in the Skalka Reservoir, showing varying depths and distances to thalweg and dam.

An important stratigraphic feature of the retrieved sediment cores was the interface of pre-dam river sediments and finer, organic matter- and Fe-enriched reservoir sedi-

ments, which are typical of central European reservoirs [20–22,28,29]. Time constraints for sediments were obtained using gamma spectrometry.

### 2.3. Sampling and Processing SPM and Plants

SPM is mostly discussed below because it best represents the current contamination status of the river systems [30]. Historical SPM sampling and water sampling were performed at sites 9825 and 1105, respectively (Figure 1) in the Röslau River near the state boundary; it was started in the 1970s by Povodí Ohře (POH), Czech state enterprise. During that time, SPM samplers were not developed/available. Thus, a car steel disc, fixed on the river bottom called 'bantam' was used by POH to continuously collect SPM for the entire period of time shown in Figure 3. SPM samples were sieved to <63 μm prior to Hg determination.

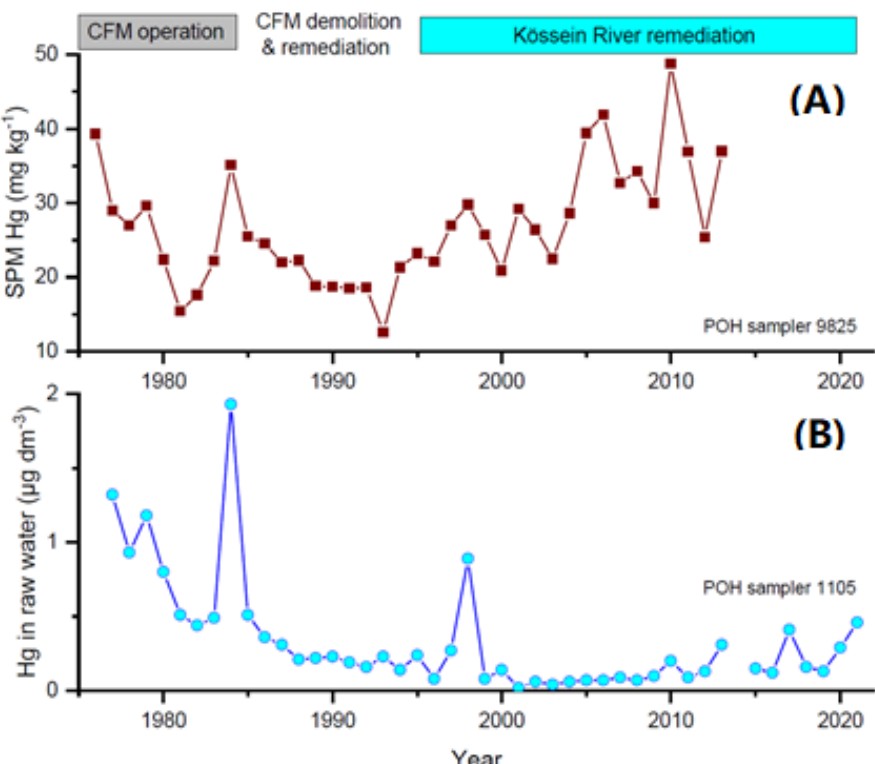

**Figure 3.** Temporal development of Hg concentrations in SPM (**A**) and unfiltered water (**B**) from the Röslau River near the state boundary. Annual averages are shown, positions of sampling sites are shown in Figure 1.

The original bantam device was destroyed by floods in 2013, and subsequent sampling was performed using other types of samplers.

In 2021 and 2022, SPM was also sampled along the Kössein–Rösslau–Ohře river system (sampling sites are shown in Figure 1) to determine where Hg entered the river particles. Each campaign included sampling at 6 to 11 sites with increasing river km from the CFM. Reliability of SPM quantification from was checked by duplicate sampling at two sites. SPM was obtained by centrifugation (4000 rpm for 15 min) of 1 dm$^3$ of river water; SPM was cooled to $-18$ °C and then freeze-dried at $-52$ °C for 2 d. The dry matter was weighed, and the entire portion was analysed using the AMA-254 (see below).

In October 2021, aquatic plants were sampled along the Kössein–Rösslau–Ohře river system in a similar manner to that of SPM. Some submerged macrophytes are known to accumulate Hg from contaminated aquatic environments [31–33] and thus represent a gate to the food web. Samples of plants without roots (leaves and stems) were collected in plastic bags, cooled, transported to the laboratory, thoroughly washed with distilled water, cooled to $-18$ °C, and freeze-dried at $-52$ °C for 2 d.

### 2.4. Chemical Analyses

Since the 1970s, historical analyses were not performed under standardised laboratory protocols because they did not exist before the 1990s. The total Hg in raw water was analysed using unfiltered river water from the sampling site 1105 (Figure 1).

The total Hg content was determined using cold-vapour atomic spectrometry after sample treatment with $SnCl_2$ using a procedure developed in the Czech Republic and later employed in a commercially available device, the AMA-254 (Altec, Czech Republic, now produced by LECO Korea Ltd., Seoul, Korea.). Analyses of the Skalka Reservoir sediments and SPM and plants used in 2021 and 2022 were also analysed using the AMA-254.

Other elements in the Skalka Reservoir sediments were determined by X-ray fluorescence spectroscopy as described by Álvarez-Vázquez et al. [27]. The Al/Si ratio was used as a proxy for sediment size as it was substantiated and verified in preceding studies on the Skalka Reservoir sediments [27,34].

### 2.5. Gamma Spectrometry and Ways to Time Constraints

Gamma spectrometry was performed using a liquid N2-cooled HPGe detector GCW2022-DET (Mirion Technologies, Canberra, Australia) with an efficiency of at least 20%. The gamma activity of the 2.5 cm inner layer of Pb shielding was <5 Bq kg$^{-1}$. Spectra were evaluated using the LYNX-MCA software (Mirion Technologies, Canberra). The detector had a suitable geometry for the analysis of small samples. The typical sample size was 2 g, and glass vials were used as the measuring cells. The samples were kept in vials for at least 3 weeks before analysis. This setting was not optimal from the point of view of measurement statistics, however, it was dictated by the need to analyse the limited size of samples from gravity cores. Therefore, subsamples from short (2–3 cm thick) depth segments of the sediment gravity cores (diameter of 5 cm) were analysed. Two reference materials were used for calibration, $^{137}$Cs and 210Pb in the same vials as those used for analyses of sediments, both provided by Czech Metrology Institute (Prague, Czech Republic).

The $^{137}$Cs activity was determined from the spectral line at 661.7 keV. The background signal from the vials was below the detection limit of the detector. Determination of unsupported 210Pb activity (further labelled 210Pb*; it is also denoted excess activity) in sediments has been recommended to confirm sediment time constraints by $^{137}$Cs [35]. $^{210}$Pb* was obtained by subtracting the activity peak at 46.5 keV and the activity peak of $^{214}$Pb at 77.11 keV, both after the subtraction of the corresponding background values for empty cells. The calibration of energy efficiency was improved by comparing $^{210}$Pb and $^{214}$Pb signals in the Chrudimka River sediments buried by the Seč Reservoir sediments and older than 90 years; their sedimentary environments have been described by Matys Grygar et al. [20] and Sedláček et al. [29].

The Chernobyl nuclear accident on 26 April 1986 spread radioactive contamination to waste areas in Europe. The average soil contamination in the Czech Republic was ca. 4000 Bq m$^{-2}$ [36]. Before the accident, a smaller $^{137}$Cs activity peak occurred in the first half of the 1960s due to the climax of nuclear weapon testing in the atmosphere during the Cold War, coeval with the construction of the Skalka Reservoir. The activity of $^{137}$Cs from the Chernobyl peak depended on the depositional system settings as well as on the geographical position in central Europe due to spatially uneven Chernobyl fallouts. Thus, the values of the Chernobyl $^{137}$Cs maxima in the Czech Republic varied, e.g., 200–400 Bq kg$^{-1}$ in the Brno Reservoir [37], ca. 100 Bq kg$^{-1}$ in the Nové Mlýny Reservoir [28], and 100–200 Bq kg$^{-1}$ in the Seč and Křižanovice reservoirs in the Chrudimka River [29]. Larger activities can be expected in the Skalka Reservoir than those in the Chrudimka River reservoirs and Nové Mlýny Reservoir in Moravia based on the Chernobyl fallout map for the Czech Republic, according to a map provided by the Czech State Authority for Nuclear Safety [36].

Ivanov et al. [35] discussed an upward decrease in $^{137}$Cs activity above the Chernobyl peak in the Schekino Reservoir, Russia, with a 10 m high dam, i.e., with the depth of

water column comparable to the Skalka Reservoir. The Chernobyl peak in the Schekino Reservoir was followed by a fast and smooth upward decrease in $^{137}$Cs activity, consistent with the sequestration of the Chernobyl accident fallout in the reservoir catchment [35]. Reworking of $^{137}$Cs-enriched layers into younger layers may occur as a consequence of floods, underwater slides, and proximal sediment reworking [29], which tends to decrease the maximum $^{137}$Cs activity by dilution.

## 3. Results

### 3.1. Mercury Concentrations in SPM and River Water

The results of historical Hg monitoring in SPM and the total dissolved Hg content from the 1970s are shown in Figure 3 in the form of annual averages based on 3–11 (SPM) and 7–20 (dissolved) samples per year. The major strength of these records was the use of the same methodology for each series, thus providing an estimate of temporal changes in the contamination status. The SPM sampling series was terminated by the destruction of the bantam sampling device during the 2013 flood. SPM sampling was then renewed by other samplers and could not be directly appended to the historical series. The SPM Hg contamination increased continuously from approximately 1993, which broadly corresponded with the end of the CFM remediation (Figure 3). De-silting of "reservoirs" at Kössein and Röslau Rivers [5], probably smaller in-channel water retention bodies might have caused a final wave of contaminated SPM that reached the Skalka Reservoir with a certain time lag. Dissolved Hg concentrations reacted more rapidly (peaking in the late 1990s) and soon returned to their low levels.

In 2021, longitudinal (downstream) SPM sampling was performed in the Kössein–Röslau–Ohře river system in three sampling campaigns (Figure 4). The Hg content in SPM did not increase immediately downstream of the CFM, but it was lower in the Kössein River and then it grew. Further downstream, the SPM Hg contamination level did not decrease, showing steady secondary contamination or lack of dilution by less contaminated sediments.

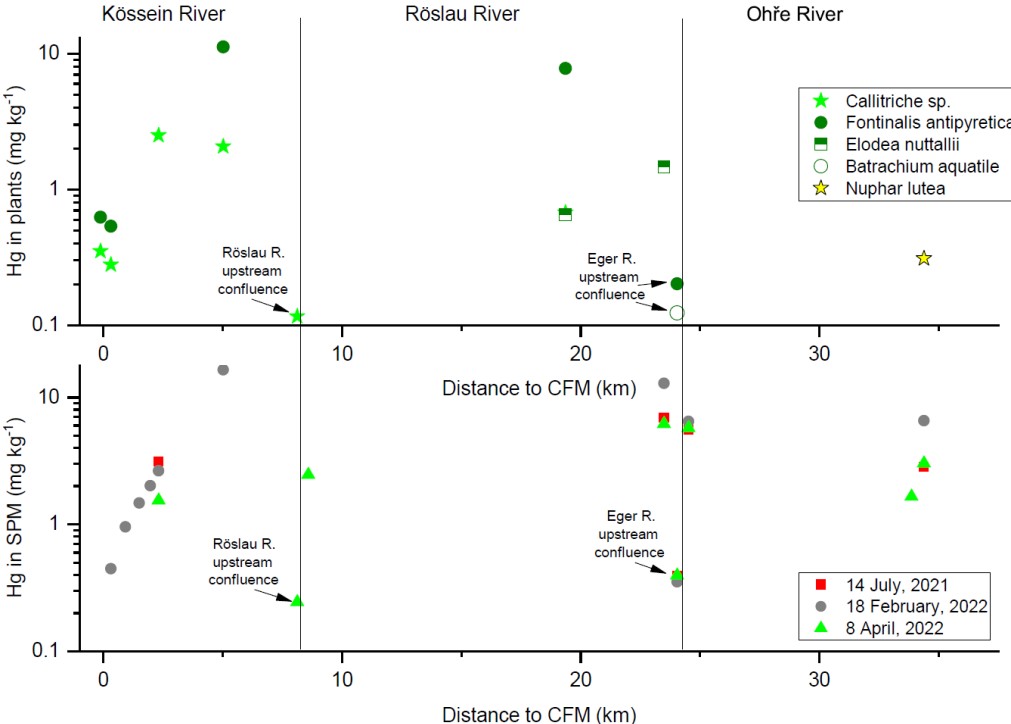

**Figure 4.** Longitudinal trends of Hg concentrations in aquatic plants (October 2021) and in SPM sampled during the three episodic events in summer 2021 and spring 2022.

### 3.2. Skalka Reservoir Sedimentary Archive

The major features of the reservoir sediment depth profiles are summarised in Table 1. Selected examples of the depth profiles are shown in Figure 5 and Supplementary Materials.

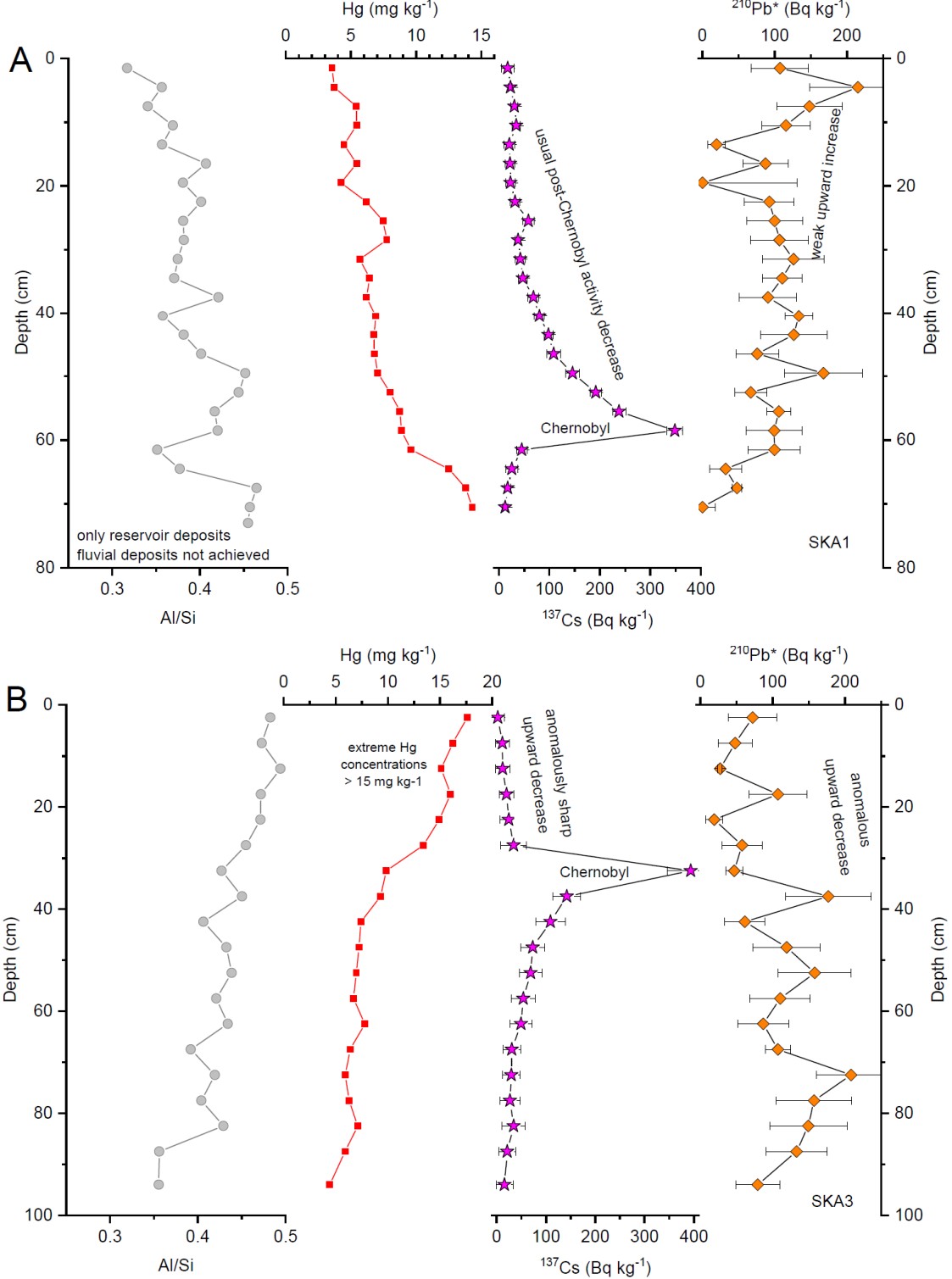

**Figure 5.** SKA1 (**A**) and SKA3 (**B**) cores from the Skalka Reservoir. Al/Si is proxy of sediment fineness, Chernobyl $^{137}$Cs peak is used as a key stratigraphic marker. Other profiles are shown in Supplementary Materials.

**Table 1.** Characteristics of the Skalka Reservoir sediment cores.

| | Water Depth (m) | Position of Core | Core Length (cm) | Fluvial Base (cm) | $^{137}$Cs Peak (cm) [(Bq kg$^{-1}$)] | Hg in $^{137}$Cs Peak (mg kg$^{-1}$) | Hg below $^{137}$Cs Peak (mg kg$^{-1}$) | Hg above $^{137}$Cs Peak (mg kg$^{-1}$) | $^{210}$Pb* above $^{137}$Cs Peak | $^{137}$Cs above $^{137}$Cs Peak |
|---|---|---|---|---|---|---|---|---|---|---|
| SKA1 | 8.9 | Near dam | 72 | No | 59 [348] | 9 | 12–14 | 4–8 | Variable | Decline |
| SKA3 | 8.4 | In thalweg | 98 | No | 32.5 [394] | 9.8 | 6–8 | 13–18 | Low | Low |
| SKA4 | 7.5 | In thalweg | 82 | 78 | 13 [193] | 10 | 6–8 | 13–14 | Increase | Low |
| SKA6 | 6.0 | Basin | 36 | 20 | 20 [79] | 8.7 | 2 | 7–9 | Increase | Slow decline |
| SKA9 | 5.1 | Basin | 65 | 38 | 25 [180] | 12.8 | 8–11 | 18 | Low | low |

The pre-dam deposits by the river retrieved in SKA6 and SKA9 contained ca. 2 and 8 mg kg$^{-1}$ Hg, respectively. Between dam constructions and the Chernobyl accident, Hg concentrations were still ca. 10 mg kg$^{-1}$ Hg at SKA6, and they remained so during the Chernobyl $^{137}$Cs peak. The strata above the Chernobyl peak were of two distinct types: (1) with Hg increase to 13–18 mg kg$^{-1}$ or (2) with stepwise declining Hg concentrations (SKA1, SKA6). In case (1), the sediments above the Chernobyl peak had very low gamma activities of $^{210}$Pb*, while in case (2), $^{210}$Pb* had an increasing activity and $^{137}$Cs had a stepwise declining activity. Case (2) was consistent with the stepwise washing of Chernobyl $^{137}$Cs fallout from the catchment and younger sediments deposited in a regular stratigraphic order. Such an evolution was found in similar reservoir sediment profiles, that is, with post-Chernobyl $^{137}$Cs activities slowly decreasing from the peak, but still with larger $^{137}$Cs activities than those of the pre-Chernobyl ones [35]. In case (1), the $^{210}$Pb and $^{137}$Cs poor material in SKA3 and SKA9 above the Chernobyl $^{137}$Cs peak was coupled with the highest Hg deposits, which could result from the re-deposition of older Hg-rich sediment material.

Sediment grain sizes, which ranged mostly from sand to silt, were the major textural features of the river and reservoir deposits analysed in our study. Figure 6 was plotted to evaluate the possibility that the Hg concentration patterns resulted from the textural changes in sediments above the Chernobyl $^{137}$Cs peak. The Hg concentration was the highest in the finest (silty) sediments (with high Al/Si, as proven by Álvarez-Vázquez et al. [27] and Talská et al. [34]) and thus, fine sediments were most relevant for the total amount of Hg contamination in the reservoir bottom. At the same (high) Al/Si ratio, sediments deposited in the 1990s are more contaminated than in 1986, indicating that textural changes in sediments did not drive the post-Chernobyl Hg maximum.

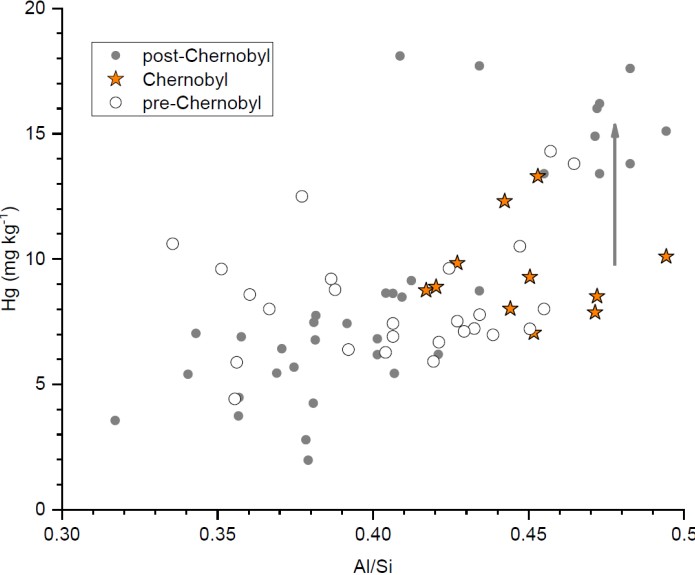

**Figure 6.** Dependence of Hg concentration on Al/Si ratio. Gray arrow shows the main trends above the Chernobyl peak.

### 3.3. Hg Concentrations in Aquatic Plants

Figure 4 shows Hg concentrations in the stems and leaves of hydrophytes in the Kössein–Röslau–Ohře river system. The aquatic plant assemblages change in the downstream direction; thus, it is not possible to construct downstream trends for a single species. Results showed elevated Hg concentrations in plants in the middle and lower parts of the studied river segment with little downstream trend, similar to that of the SPM (Figure 4).

## 4. Discussion

### 4.1. Timing of Sediment Contamination Climax

Since the dam construction in the 1960s, the Skalka Reservoir recorded the contamination of the Kössein–Röslau–Ohře river system. The sediment stratigraphy of the Skalka Reservoir is complex because of its small depth and seasonal manipulation of water levels [2,23,27]. The Skalka Reservoir water level has occasionally decreased to a point where half of the bottom area is exposed [9]. Pacina et al. [38] showed that sediment reworking in the inflow to the Skalka Reservoir resulted from variations in the water level. Therefore, manipulation of the water level could redistribute the Hg-rich layers to downstream and more distal sites by so-called sediment focusing [39]. Thus, we did not construct a depth-age model based on the assumption of continuity of sediment deposition in each of the studied cores, but we considered the possibility of hiatuses in the sediments. Here, we followed the recommendation of Miller et al. [40] to produce reliable and unbiased conclusions on historical contamination. For these fluxes of Hg or radionuclides to the reservoir cannot be calculated.

$^{137}$Cs Chernobyl peak values increased downstream, with the highest values near the dam, which was probably related to lithological changes because of the fine-grained nature of distal sediments. A similar pattern has been observed in other Czech reservoirs [29]. $^{137}$Cs is known to bind to fine and clay particles [41,42], similar to most secondary contamination by risk elements.

The following conclusions were drawn from the Skalka Reservoir sediments (Figure 5 and Supplementary Materials). The Hg concentration in the Chernobyl $^{137}$Cs peak, which was only one year after closure of the CFM, was ca. 10 mg kg$^{-1}$, slightly more than the Hg concentrations deposited in the 1960s between dam construction and the Chernobyl peak (Table 1). In two cores (SKA1 and SKA6), the Hg concentration above the Chernobyl peak decreased; however, in three core sediments (SKA3, SKA4, SKA9), a stratum was found with anomalously low $^{210}$Pb* and $^{137}$Cs activities and extremely high Hg concentrations of ca. 15 mg kg$^{-1}$. In these cores, the post-Chernobyl $^{137}$Cs activity was lower than that in the pre-Chernobyl state, which was just the reverse to the records under continuous deposition of material from the reservoir catchment surface with the Chernobyl dust fallout [35]. Thus, the Hg-contaminated sediment was deposited in some parts of the Skalka Reservoir after the closure of the factory; maximum concentration of Hg was found in strata with low fallout radionuclide activities, with older sediments originating upstream of the reservoir.

The peak Hg concentration readings after the Chernobyl peak in SKA3 and SKA9 were in agreement with the increase in SPM Hg concentration in the lower Röslau River in the 1990s and the 2000s (Figure 3). Maršálek et al. [9] and Paulin et al. [10] attributed the increase in contamination of rivers to the dredging of the Kössein River bottom and reservoir deposits around 1996. Pedall et al. [5] also assumed that the extraction of contaminated bottom sediments from the Kössein River enhanced secondary contamination in the 1990s.

The reason for the increase In Hg concentration in SPM in the lower reach of the Röslau River in the 1990s during the extraction of the Kössein River bottom deposits and in the subsequent decades remains unknown (Figure 3). Pedall et al. [5] identified bank erosion in freely flowing Kössein River segments as a major source of secondary contamination in the 2000s. They estimated that the total amount of material mobilised by lateral bank erosion contained 420 kg Hg annually; the majority of this material had been deposited in floodplains during floods [5]. Pedall et al. [5] recommended the renewal of stone pavements at the river banks (Section 6.1.2 in the quoted report), although keeping old embankments

is against the basic WFD concept. The embankments have not been renewed although they have been damaged in a stepwise manner by floods [5,10].

In our previous study [2,26] we assumed that the climax of sediment contamination soon followed the onset of contamination, as it was common for early technologies with low efficiency and low environmental concerns. This conclusion was based on overbank deposits in the Röslau River floodplain. However, overbank deposits can only record contamination history when maximum contamination enters the river system at high discharges with overbank floods. Such a case occurs, for example, after a flush of contamination from primary sources by extreme precipitation from a failure of settling ponds with mine waste [43,44], which was not the case in the river systems downstream of the CFM. Conversely, if maximum contamination enters the river system under low river discharges, for example, from wastewater effluents, it can leave no record in overbank fines, which was the case in the studied river system. Therefore, the Skalka Reservoir sediments were preferred as medium recording historical contamination in this paper.

### 4.2. Current Contamination Status and Its Causes

The maximum Hg concentration in aquatic plants from the Kössein–Röslau river system collected in October 2021 was more than $10$ mg kg$^{-1}$ (Figure 4). Cosio et al. [31] and Bonanno et al. [32] found biogenic accumulation of Hg: they found Hg in the range of $0.1$–$2$ mg kg$^{-1}$ Hg in submerged aquatic plants. It is thus not surprising that mercury concentrations in aquatic plants in the Kössein–Röslau river system were so high. Because of the known bioaccumulation of Hg in the food web, it became evident why fishes in these rivers and the Skalka Reservoir were not recommended for consumption. In 2010, 73% of the fishes in the Kössein–Röslau river system exceeded the safety limits for consumption [5]. Because Hg in Kössein-River channel sediments is bioavailable to aquatic plants (Figure 4), their entry into aquatic organisms is inevitable.

A downstream trend (Figure 4) showed that Hg concentrations in SPM increased downstream of the Kössein River to their maxima at ca. $10$ mg kg$^{-1}$ and remained elevated (though scattered) in the Röslau River. Therefore, the sources of this contamination could be the bottom sediments and banks of the Kössein River. In 2010, the Kössein-Röslau channel sediments showed a mean concentration of $32$ mg kg$^{-1}$, and similar Hg concentrations were found in contemporary SPM [5]. The Hg contamination of SPM has decreased to ca. $1/3$ in the last decade, which was a positive trend.

### 4.3. Priorities and Pitfalls in River Revitalisation

Historical contamination can endanger the lives of people close to contamination hotspots [1,3]. Rivers in industrialised areas have already received high volumes of contamination before they were regulated, straightened, and frequently canalised or paved by stone or concrete. Past river engineering has prioritised technological and hydrological aspects [45,46], which is currently perceived as a problem due to decreased channel-floodplain connectivity, absence of wetland habitats for biota, and an unwanted state of river landscapes [46]. Currently, there is a preference to liberate rivers from former concrete and stone-walled channels and re-establish their semi-natural littoral habitats [45]. However, replacing past engineering paradigms with re-naturalisation paradigms is questionable when legacy contamination is involved and rivers are left to remobilise the historical burden in the proximal floodplain [5,6].

Currently, in European river management, there is no need to include historical contamination in planning revitalisation or 'renaturation' of water systems. The WFD declared a preference for the status of natural rivers, that is, leaving water systems to their natural behaviour. The WFD has been implemented more systematically in Bavaria than in other Central European countries. In the case of Kössein and Röslau rivers, the obvious solution to stop the spread of legacy contamination further downstream of contamination hotspots would be to prevent bank erosion, similar to the recommendation of Ciszewski et al. [6] for a creek draining historical ore mine in Poland. The 'renaturation' of the Kössein

River banks would not be recommended, as noted by Pedall et al. [5]. The Hg concentration in the littoral zone of the Kössein-Röslau Rivers is in the order of hundreds of mg kg$^{-1}$, with maxima up to 300 mg kg$^{-1}$ [2,5,10]. Any manipulation with banks other than their sealing to prevent erosion is risky and should be performed with extreme caution to the downstream river system.

Pedall et al. [5] recommended stabilising the Kössein and Röslau banks using stone pavements, although this contradicted the aims of the WFD. Paulin et al. [10] have concluded that there are two possible options for managing the littoral zones of these water systems: (1) extraction of contaminated sediments from floodplains or (2) complete prevention of bank erosion [10]. Solution 1 would not be economically feasible, according to Pedall et al. [5]. Wherever the banks of severely contaminated rivers are sealed, they should remain like that instead of being liberated [5,6]. Contaminated channel banks will otherwise become a persistent source of contamination by both physical erosion and chemical mobilisation and risk element migration by subsurface flow, even without physical remobilisation [43,47].

In the Czech Republic, there was no obligation to perform a risk assessment of secondary contamination before river revitalisation. Tůmová et al. [48] showed an example of the construction of an artificial wetland in a floodplain of the Panenský Creek, which historically served as a disposal site of waste from lead glass cutting; this 'renaturation' resulted in secondary contamination of the river downstream [48,49]. Several 'renaturation' projects were also planned for the nearby Ploučnice River, in particular re-activation of its formerly meandering channel [50], although it has been severely contaminated because of failing settling ponds containing mine waste in 1981 [44]. These 'renaturation' projects have been halted because of personal responsibility of the architects of the project and the unwillingness of the landowners to let the river laterally erode their plots. The only existing limit for the revitalisation of contamination hotspots is the possible disposal of sediments extracted from the revitalised areas; however, there are no limits for the reuse of extracted contaminated material in the river system. Thus, there are no limits to the spread of contamination downstream of the river system.

Domínguez et al. [43] and Hornberger et al. [51] concluded that the success of revitalisation of severely contaminated river systems must be evaluated at the scale of the entire river system connected to the contamination hotspot. In this sense, the revitalisation of the Kössein-Röslau river systems in the 1990s and the 2000s likely solved some problems in those river stretches very close to the CFM, but harmed the floodplains downstream [5]. This particular case can serve as an example of the insufficiency of the WFD for the sustainable management of historically impacted water systems.

## 5. Conclusions

The sedimentary archive of the Skalka Reservoir was analysed, time constrained using gamma spectrometry, and compared with the historical record of Hg concentrations in suspended particulate matter in the lowermost Röslau River. The export of the Hg-contaminated river particles from the CFM and the Kössein River peaked in the mid-1990s and in the 2000s, during and soon after remediation and revitalisation of the abandoned factory and the Kössein River channel, when river SPM contained ca. 30 mg kg$^{-1}$ Hg and the Skalka Reservoir had deposits of up to 18 mg kg$^{-1}$ Hg. The SPM in the Röslau River and aquatic plants in the river bottom contained ca. 10 mg kg$^{-1}$ Hg, which exceeded the global averages by two orders of magnitude and endangered aquatic life. This secondary contamination is persistently transported from the German-Czech boundary to the Skalka Reservoir and the Ohře River system. This case study documented the need to establish safety frames for the management of contaminated river systems, the revitalisation of which should prevent enhanced contamination of downstream river systems. Currently, there are no safety limits for the revitalisation of contaminated rivers, intervention thresholds, or thresholds for SPM exported from contaminated rivers downstream. Renaturation of anthropogenic rivers, one of the aims of the WFD, should be performed with caution, and

secondary contamination should be prevented. Corresponding legislative frames should be urgently prepared for managing historically impacted rivers to protect current and future river systems.

**Supplementary Materials:** The following supporting information can be downloaded at: https://www.mdpi.com/article/10.3390/w14213481/s1; Figure S1: Depth profiles of core SKA4; Figure S2: Depth profiles of core SKA6; Figure S3: Depth profiles of core SKA9.

**Author Contributions:** Conceptualisation, T.M.G. and T.N.; methodology, T.M.G., T.N. and J.S.; formal analysis, J.R.; investigation, T.M.G., M.H., T.N. and J.S.; resources, J.B. and J.H.; writing—original draft preparation, T.M.G.; writing—review and editing, T.M.G. and O.S.; visualisation, T.M.G., J.P. and J.E.; supervision, T.M.G.; project administration, T.M.G., T.N. and J.E.; funding acquisition, T.M.G. and M.H. All authors have read and agreed to the published version of the manuscript.

**Funding:** The work was supported by the Czech Science Foundation, project numbers 20-06728S (sampling in 2021 and 2022, gamma spectrometry, Hg analysis, and data evaluation) and 17-06229S (Skalka Reservoir sediment sampling).

**Data Availability Statement:** Not applicable.

**Acknowledgments:** We acknowledged laboratory analyses by M. Maříková and P. Vorm (IIC CAS Řež) and M. Roll (GlI CAS Prague). The work was inspired by L. Majerová (Czech Inspection of Environment, Ústí nad Labem).

**Conflicts of Interest:** The authors declare no conflict of interest.

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
