# Peer review of "Lessons Learnt from the Revitalisation of Chemical Factory in Marktredwitz and River Banks Downstream: When ‘Renaturation’ Can Be Harmful"

_water, doi:10.3390/w14213481_

Round 1

Reviewer 1 Report

Dear Editor,

I read the paper by Matys Grygar et al., and I found the subject extremely interesting. The environmental concerns posted by the authors are very valid and the potential risk from the revitalization of a chemical factory in Marktredwitz is of great concern. However, I found the paper very weak in terms of real evidence and the story can be improved significantly by clarifying, adding more details as well as additional analysis. At the moment, I find the paper very speculative. I am missing statistical analysis to support some of the authors' conclusions. In addition, the authors should make age models based on the 210Pb and 137 Cs signals and if possible convert everything to fluxes for better comparison. I am also missing details in the sample campaigns (e.g., how many samples were taken?).  I think is quite uncommon to explain some of the trends observed in the cores the way the authors are doing it. In other words, why do the authors always refer to Chernobyl (1986), when they can derive sedimentation rates? Also, with the small size of grab samples, I am not sure if many of the discussions by the authors can be inferred.

I believed, the authors have a lot of good data and potential for a nice paper, however, the quality of the paper needs to be significantly improved.

Author Response

Reviewer 1

Dear Editor,

I read the paper by Matys Grygar et al., and I found the subject extremely interesting. The environmental concerns posted by the authors are very valid and the potential risk from the revitalization of a chemical factory in Marktredwitz is of great concern. However, I found the paper very weak in terms of real evidence and the story can be improved significantly by clarifying, adding more details as well as additional analysis. At the moment, I find the paper very speculative. I am missing statistical analysis to support some of the authors' conclusions. In addition, the authors should make age models based on the 210Pb and 137 Cs signals and if possible convert everything to fluxes for better comparison. I am also missing details in the sample campaigns (e.g., how many samples were taken?).  I think is quite uncommon to explain some of the trends observed in the cores the way the authors are doing it. In other words, why do the authors always refer to Chernobyl (1986), when they can derive sedimentation rates? Also, with the small size of grab samples, I am not sure if many of the discussions by the authors can be inferred.

The manuscript goal is evaluation of remediation activities, for which timing of contamination climax is of key importance. We did not plan to completely describe the gamma activity distribution in sediment cores/strata. Meaningful fluxes cannot be calculated without perfect stratigraphic order, somehow stable sedimentation rates, stable activity of radionuclides in sediment source etc. Deposition in reservoirs have more or less regular component as well as episodic components related to water level manipulation and floods. There is no need to work with fluxes to identify pre-Chernobyl, near Chernobyl, and post-Chernobyl parts of the sediment record, as actually the post-Chernobyl years are of key importance for the study – CFM and river system remediation were performed soon after the Chernobyl peak. We desired to get robust time constraints for our conclusions, and we thus agree with the Reviewer 1 our way of data presentation is “uncommon”, but we do not think it is speculative. What we are concerned about is that it would be the weakly based construction of age model and the relevant calculation of sediment fluxes would be speculative. We added one sentence to the 1st paragraph of Discussion that it is not possible to convert results to fluxes into the revised version of the manuscript (after text summarising the sediment stratigraphy).

We did not use grab sampling of reservoir sediments; gravity cores were used. It is common way of reservoir bottom sediment sampling; small size is not a problem – it is used conventionally in such studies.

The quality of SPM sampling in 2021 and 2022 (pumping water and its centrifugation) was tested by comparing duplicate samples at several sites (not shown). This way we assessed SPM, which gets transported without any doubts. It indicates that the origin of the SPM carrying the highest Hg contamination is far below the CFM. Our discussion is further supported by the aquatic plants showing the similar trends.

Reviewer 2 Report

The manuscript by Matys Grygar et al. deals with an important problem of mercury heritage contamination with transboundary implications. After a considerable cleaning action at the industrial site in Marktredwitz (Germany) in the 1990s, severe mercury contamination of rivers and downstream reservoir increased. New observations, that complete previously reported data, concern the concentrations of Hg in riverine suspended particles and aquatic plants collected in 2021 and the concentrations recorded in sediment cores recovered from the Skalka Reservoir (Czech Republic). The authors concluded that river "renaturation" projects in the frame of the Natura 2000 regions should carefully examine their environmental pertinence to avoid enhanced erosion of contaminated riverbanks, which may further increase secondary contamination downstream.

The manuscript present original data, previously reported data (which are not easily comparable) and opinions concerning pertinence of "renaturation plans".  A better organisation of these materials between introduction, discussion and conclusions with some logical links between obtained results and conclusions would clearly improve the quality of the paper. Although the general conclusions based on this study and previously published data seem to be well founded, several problems related to the methods, interpretation of results and manuscript structure need clarification.

1.    All samples of riverine suspended particulate matter (SPM) and aquatic plants showed high concentrations of Hg, increasing along the first several kilometres downstream from the factory site and then remaining fairly constant, even in the samples from the Skalka Reservoir outflow (Ohre River). This is somewhat surprising, because the SPM from the Röslau River and Eger River upstream from confluences shows "near-background" Hg concentrations. No data are given on SPM loads of these rivers, but one would expect a decrease of Hg concentrations by dilution. Could you comment?

2.    SPM sampling in this study (instantaneous sampling, centrifuging) and previous studies (integrative sampling) are very different. Moreover, the latter method is described in different ways in the present manuscript and by Hosek et al. 2020. Please, compare lines 156-161 in this manuscript with Hosek et al. 2020: "For SPM sampling, POH historically used metallic samplers attached to the river bottom, and since the 1990s, they were replaced by floating SPM samplers called ‘‘bantams’’ similar to those used in this study". 
Are the comparisons in lines 380-383 pertinent? Please, explain and comment on the pertinence of comparisons with centrifugation method.

3.    In the method section, please, add detection limits of measured radionuclides at the counting times used in this study, provide information on counting errors of 210Pb, activities, counting errors of 214Pb and propagated errors of unsupported 210Pb. The later errors (at 1 sigma) should be plotted in figure 5 and in figures in Supplementary Materials. Also explain in the method section how sediments were subsampled (slice thickness, depth intervals).

4.    Profiles of 137Cs, 210Pbx excess and Hg in sediment cores are difficult to interpret because of the variable sediment grain-size (approximated with Al/Si), possible hiatuses due to sediment erosion and redeposition. Thus no dating with 210Pb was attempted. Proposed interpretation is reasonably cautious, except for core SKA6 where no clear 137Cs peak is present (small increase at 20 cm depth is probably due to the increase in fine particles). There is no attempt to relate the observed variations in radionuclide profiles with sediment lithology (colour, structure, texture, water content). In figure 5, in core SKA 1,  210Pb excess activities show strong variations (from close to 0 at about 13 cm depth to >200 Bq kg-1 at about 4 cm depth), while 137Cs activities present a smooth profile (and nearly the same activities at 13 and 4 cm depths). How it can be explained? 

5.    Hg concentrations in sediments in each core tend to increase with increasing Al/Si ratio and a broad correlation is shown in fig. 6. The regression line and determination coefficient (R2) could be useful. However the meaning of two arrows is not well explained. I guess, the idea is that at the same (high) Al/Si ratio, sediments deposited in the 1990thare more contaminated than in 1986 and at low Al/Si ratio post-Chernobyl Hg concentrations are lower than pre-Chernobyl sediments?  Right? But why? Please, reformulate lines 290-293. 

Comments on the figures: 

In general, the figures are not good quality and the figure captions must be revised. 

Fig 1. Remove all sampling sites names in the panel B, which are not used in this study. If used, explain their meaning in figure caption. 

Fig. 2. Remove the coring site not discussed in this work (SKA 2) or explain why it was not used.

Fig. 3. Figure caption indicates that panel B shows Hg concentration in unfiltered water, but Y axis in fig. 3 reads "Hg dissolved". It is not the same (dissolved by convention is in water filtered at 0.45μm). Figure caption should refer to the sampling site(s) shown in fig. 1 (9825 or/and? 1105) and refer to data sources (POH?). Although announced in the figure caption, letters A and B are not show in figure 3. 

Fig. 4. The caption announces "SPM sampled in single days in summer 2021", but in the legend there are samples also from February and April 2022. Hg concentrations in plants are shown but nothing is mentioned in the figure caption (units refers to plant dry weight?, what part of plant was examined? Root? Shoots?). This should also be explained in the method section. It seems that sampling sites on Röslau and Eger Rivers (upstream from confluences) are not on the map in fig 1.

Detailed comments on the text:

Line 24. The sediments were not datedwith unsupported 210Pb. The same applies to similar statements in the whole text.

Lines 85-87. Not really. Atmospheric emission are mainly recorded as Hg0 and may depend on the production intensity or dispersion in local soil, while emissions to aquatic system depend on the liquid waste discharges related to the production, leaks or deliberate elimination.

Line 134. Add the dates of sampling. 

Line 163. It seems that sampling points indicated in Fig. 1 are not (or not all) the same as in Fig. 4.

Line 180. Unfiltered water measurements normally provide the total Hg concentrations in raw water (not dissolved Hg). But what really is measured in raw water depends of specific analytical protocol. 

Line 207. Unsupported 210Pb activity is usually called excess activity (210Pbex)

Line 237. "total dissolved" – see note to the line 180

Line 245. "the reservoirs at Kössein and Röslau Rivers" – these reservoirs were not mentioned before 

Lines 250-253. Statement is not clear. Please, rephrase

Lines 271-272. I do not see a clear Chernobyl peak in this core

Line 300. This is the first time we learn that Hg was measured in stems and leaves (not roots) of plant. This should be placed in sampling and method sections. I suggest introducing this paragraph just after the results of SPM. 

Line 327. Confusing. Change to "between dam construction in the 1960thand Chernobyl peak (1986)."

Lines 333-336. Do you mean displacement inside the reservoir? What mechanism can be proposed for displacement of these sediments (sliding, density currents?)

Lines 353-364. Note that in the previous paper (ref . 2 , fig. 4) you showed that high Hg concentrations in SPM were observed during high water discharge periods. The high concentrations of Hg due to wastewater input at low river flow are probably not present since the industrial site clean up.

Line 369-370. Bioaccumulation factor has no units. 

Author Response

Reviewer 2

The manuscript by Matys Grygar et al. deals with an important problem of mercury heritage contamination with transboundary implications. After a considerable cleaning action at the industrial site in Marktredwitz (Germany) in the 1990s, severe mercury contamination of rivers and downstream reservoir increased. New observations, that complete previously reported data, concern the concentrations of Hg in riverine suspended particles and aquatic plants collected in 2021 and the concentrations recorded in sediment cores recovered from the Skalka Reservoir (Czech Republic). The authors concluded that river "renaturation" projects in the frame of the Natura 2000 regions should carefully examine their environmental pertinence to avoid enhanced erosion of contaminated riverbanks, which may further increase secondary contamination downstream.

The manuscript present original data, previously reported data (which are not easily comparable) and opinions concerning pertinence of "renaturation plans".  A better organisation of these materials between introduction, discussion and conclusions with some logical links between obtained results and conclusions would clearly improve the quality of the paper. Although the general conclusions based on this study and previously published data seem to be well founded, several problems related to the methods, interpretation of results and manuscript structure need clarification.

  1. All samples of riverine suspended particulate matter (SPM) and aquatic plants showed high concentrations of Hg, increasing along the first several kilometres downstream from the factory site and then remaining fairly constant, even in the samples from the Skalka Reservoir outflow (Ohre River). This is somewhat surprising, because the SPM from the Röslau River and Eger River upstream from confluences shows "near-background" Hg concentrations. No data are given on SPM loads of these rivers, but one would expect a decrease of Hg concentrations by dilution. Could you comment?

    RESPONSE to item 1: We do not have SPM fluxes, but the Röslau and Eger rivers has almost equal mean annual discharge at their confluence. Obvious from concentrations is the entire system addressed in the manuscript has been contaminated and dilution can only concern input from the Eger River and so the Hg concentration in SPM downstream confluence decreases.
  2. SPM sampling in this study (instantaneous sampling, centrifuging) and previous studies (integrative sampling) are very different. Moreover, the latter method is described in different ways in the present manuscript and by Hosek et al. 2020. Please, compare lines 156-161 in this manuscript with Hosek et al. 2020: "For SPM sampling, POH historically used metallic samplers attached to the river bottom, and since the 1990s, they were replaced by floating SPM samplers called ‘‘bantams’’ similar to those used in this study". 
    Are the comparisons in lines 380-383 pertinent? Please, explain and comment on the pertinence of comparisons with centrifugation method.

RESPONSE to item 2: We now present samples from the steel discs called bantams by POH. I am afraid the quoted text in Hošek et al. was not precise. POH also used more kinds of samplers in history, but in the manuscript under review we have only used one of them.
We know the results of novel SPM from centrifugation and historical ones from bantams can be different, but both sampling also have different targets – 1) temporal evolution of contamination in history (POH samplers), and 2) spatial evolution of contamination in the present time (our centrifugated riverine water samples). 

  1. In the method section, please, add detection limits of measured radionuclides at the counting times used in this study, provide information on counting errors of 210Pb, activities, counting errors of 214Pb and propagated errors of unsupported 210Pb. The later errors (at 1 sigma) should be plotted in figure 5 and in figures in Supplementary Materials. Also explain in the method section how sediments were subsampled (slice thickness, depth intervals).

RESPONSE to item 3: Done

  1. Profiles of 137Cs, 210Pbx excess and Hg in sediment cores are difficult to interpret because of the variable sediment grain-size (approximated with Al/Si), possible hiatuses due to sediment erosion and redeposition. Thus no dating with 210Pb was attempted. Proposed interpretation is reasonably cautious, except for core SKA6 where no clear 137Cs peak is present (small increase at 20 cm depth is probably due to the increase in fine particles). There is no attempt to relate the observed variations in radionuclide profiles with sediment lithology (colour, structure, texture, water content). In figure 5, in core SKA 1,  210Pb excess activities show strong variations (from close to 0 at about 13 cm depth to >200 Bq kg-1at about 4 cm depth), while 137Cs activities present a smooth profile (and nearly the same activities at 13 and 4 cm depths). How it can be explained? 

RESPONSE to item 4: “Dating” was replaced by terms like “time constraints”.
I think Chernobyl peak of 137Cs was so sharp lithologic control could not much change it. I would expect it generally if sand intercalated by clay was analysed, that was not the case.

  1. Hg concentrations in sediments in each core tend to increase with increasing Al/Si ratio and a broad correlation is shown in fig. 6. The regression line and determination coefficient (R2) could be useful. However the meaning of two arrows is not well explained. I guess, the idea is that at the same (high) Al/Si ratio, sediments deposited in the 1990thare more contaminated than in 1986 and at low Al/Si ratio post-Chernobyl Hg concentrations are lower than pre-Chernobyl sediments?  Right? But why? Please, reformulate lines 290-293. 

Response to item 5: Yes, the Reviewer understood our interpretation. Sorry for unclear formulations, we replaced our original text by the first half of the sentence written by the Reviewer. Still, we do not have explanation for the lower post-Chernobyl Hg in coarser sediments. In any case, finer sediments carry more Hg and more Hg was also found in post-Chernobyl SPM.

Comments on the figures: 

In general, the figures are not good quality and the figure captions must be revised. 

Fig 1. Remove all sampling sites names in the panel B, which are not used in this study. If used, explain their meaning in figure caption. 

RESPONSE: Done

Fig. 2. Remove the coring site not discussed in this work (SKA 2) or explain why it was not used.

RESPONSE: SKA2 was removed from figure, it was not used because the thickness of reservoir sediments in this core was too small.

Fig. 3. Figure caption indicates that panel B shows Hg concentration in unfiltered water, but Y axis in fig. 3 reads "Hg dissolved". It is not the same (dissolved by convention is in water filtered at 0.45μm). Figure caption should refer to the sampling site(s) shown in fig. 1 (9825 or/and? 1105) and refer to data sources (POH?). Although announced in the figure caption, letters A and B are not show in figure 3. 

RESPONSE: Corrected

Fig. 4. The caption announces "SPM sampled in single days in summer 2021", but in the legend there are samples also from February and April 2022. Hg concentrations in plants are shown but nothing is mentioned in the figure caption (units refers to plant dry weight?, what part of plant was examined? Root? Shoots?). This should also be explained in the method section. It seems that sampling sites on Röslau and Eger Rivers (upstream from confluences) are not on the map in fig 1.

RESPONSE: The discrepancy was corrected, sampled plant parts were specified in methods section.  

Detailed comments on the text:

Line 24. The sediments were not dated with unsupported 210Pb. The same applies to similar statements in the whole text.
RESPONSE Sorry for such oversimplification, the text was corrected

Lines 85-87. Not really. Atmospheric emission are mainly recorded as Hg0 and may depend on the production intensity or dispersion in local soil, while emissions to aquatic system depend on the liquid waste discharges related to the production, leaks or deliberate elimination. RESPONSE: I know. I wanted to state that timing of CFM pollution climax depends on environment compartment, but still, fluvial contamination climax after factory closure was unexpected.

Line 134. Add the dates of sampling. DONE

Line 163. It seems that sampling points indicated in Fig. 1 are not (or not all) the same as in Fig. 4. RESPONSE: We do not see any error, corresponding co-author checked the figures and source files.

Line 180. Unfiltered water measurements normally provide the total Hg concentrations in raw water (not dissolved Hg). But what really is measured in raw water depends of specific analytical protocol. RESPONSE: The only historical datasets were not obtained under current standard protocols, as we wrote elsewhere in the manuscript; neither better datasets, nor historical protocols do exist. The terminology was corrected.

Line 207. Unsupported 210Pb activity is usually called excess activity (210Pbex) RESPONSE We added that term along with the original one

Line 237. "total dissolved" – see note to the line 180 CORRECTED

Line 245. "the reservoirs at Kössein and Röslau Rivers" – these reservoirs were not mentioned before RESPONSE: taken from Pedall et al. [5], probably some smaller pools, the text was modified

Lines 250-253. Statement is not clear. Please, rephrase DONE

Lines 271-272. I do not see a clear Chernobyl peak in this core RESPONSE We assigned 137Cs maximum to the Chernobyl accident irrespective of its sharpness – the peak sharpness likely depends on deposition mechanism and/or post-depositional mixing; even if the peak seems to not be caught by sampling, the 137Cs maximum was understood as the closest to the Chernobyl accident.

Line 300. This is the first time we learn that Hg was measured in stems and leaves (not roots) of plant. This should be placed in sampling and method sections. I suggest introducing this paragraph just after the results of SPM. CORRECTED

Line 327. Confusing. Change to "between dam construction in the 1960thand Chernobyl peak (1986)." CORRECTED

Lines 333-336. Do you mean displacement inside the reservoir? What mechanism can be proposed for displacement of these sediments (sliding, density currents?) RESPONSE: I mean mobilisation of historical contamination peak from upstream sinks in the river channel, bank erosion (with huge Hg contents).

Lines 353-364. Note that in the previous paper (ref . 2 , fig. 4) you showed that high Hg concentrations in SPM were observed during high water discharge periods. The high concentrations of Hg due to wastewater input at low river flow are probably not present since the industrial site clean up. RESPONSE: The quoted text summarises two general mechanisms of sedimentation record of pollution climax and substantiate why the record in floodplains would not be the best choice here – and it was necessary to work with the reservoir sediments, although the reservoir is shallow and its stratigraphy is complicated. The text was modified to avoid misunderstanding.

Line 369-370. Bioaccumulation factor has no units. CORRECTED

Round 2
